# Cloud Data-Driven Intelligent Monitoring System for Interactive Smart Farming

**DOI:** 10.3390/s22176566

**Published:** 2022-08-31

**Authors:** Kristina Dineva, Tatiana Atanasova

**Affiliations:** Institute of Information and Communication Technologies—Bulgarian Academy of Sciences, Acad. G. Bonchev Str., Bl. 2, 1113 Sofia, Bulgaria

**Keywords:** smart farming, Azure cloud architecture, cloud-based data pipelines, QR tags, data visualization

## Abstract

Smart farms, as a part of high-tech agriculture, collect a huge amount of data from IoT devices about the conditions of animals, plants, and the environment. These data are most often stored locally and are not used in intelligent monitoring systems to provide opportunities for extracting meaningful knowledge for the farmers. This often leads to a sense of missed transparency, fairness, and accountability, and a lack of motivation for the majority of farmers to invest in sensor-based intelligent systems to support and improve the technological development of their farm and the decision-making process. In this paper, a data-driven intelligent monitoring system in a cloud environment is proposed. The designed architecture enables a comprehensive solution for interaction between data extraction from IoT devices, preprocessing, storage, feature engineering, modelling, and visualization. Streaming data from IoT devices to interactive live reports along with built machine learning (ML) models are included. As a result of the proposed intelligent monitoring system, the collected data and ML modelling outcomes are visualized using a powerful dynamic dashboard. The dashboard allows users to monitor various parameters across the farm and provides an accessible way to view trends, deviations, and patterns in the data. ML models are trained on the collected data and are updated periodically. The data-driven visualization enables farmers to examine, organize, and represent collected farm’s data with the goal of better serving their needs. Performance and durability tests of the system are provided. The proposed solution is a technological bridge with which farmers can easily, affordably, and understandably monitor and track the progress of their farms with easy integration into an existing IoT system.

## 1. Introduction

New technologies and farming digitization offer very productive and profitable solutions in agriculture. They are crucial to improving the sustainability and competitiveness of the sector while simplifying the daily work of farmers. There is a need to ensure that all, including small and medium farmers, have access to and benefit from technology. Smart farming means the application of data and information technology to improve the production processes of complex farming systems [1]. The development of smart farms is already moving from the “emerging” phase to the “growth” phase [2,3] in its technological lifecycle. This phase in the advancement of smart farming is seen in the emergence of data-driven solutions from remote and automated sensor systems [4]. Smart farms collect a huge amount of data from abundant sources about the environment [5] and the condition of animals [6], plants, soil, etc. Data visualization is achieved through excel tables, which contain a lot of numerical and categorical values in raw format. These data are most often stored locally and do not provide opportunities in their entire range to be processed by farmers. Usually, these kinds of tabular representations are difficult to understand, and the process of extracting some useful information from the farmer is slow and cumbersome; furthermore, sometimes it is a rather overwhelming and annoying task. This leads to a sense of the uselessness of the collected data and a lack of motivation on the part of farmers to invest in modernizing farms.

This sense of the uselessness of the collected data is in accordance with a recent Gartner survey [7] about a high percentage of organizations which experience a great degree of regret over their largest purchases related to high technology. The major cause of technology purchase regret among the buyers is the subsequent mismatch between the expected results and an unclear goal concerning what is possible to achieve with the purchased technology. According to Gartner, 67% of people involved in technology purchasing decisions are not IT professionals, and that causes issues. New cloud technologies, even for common solutions, require migration and involve a learning curve before becoming fully operational [8]. However, the way agriculture works is changing before our eyes [9].

The trend of bringing more intelligence to farm monitoring and control systems is widely observed. An intelligent system (IS) can be defined as a system that incorporates intelligence into machine-processed applications. It is an advanced system that can collect, analyze, store, and respond to the data it gathers from the environment. It can operate and communicate with users or other computer systems. It can also learn from experience and adapt according to current data [10,11].

Advances in digital technology are leading to the development of intelligent systems that can monitor [12], control, and visualize various farm operations [13] and animal status in real-time. The wireless sensor network and cloud application software management system based on the Internet of Things (IoT) are used to design an intelligent monitoring system for the facility’s agriculture environment [14]. An intelligent monitoring system which is based on machine vision, an automatic identification model, a Web client, and a cloud server are developed in [15].

The smart farming applications are IoT based and generate large amounts and high-resolution data. In [16], a study is presented about implemented sensors to detect and report cow behavior in real-time. The authors used various sensors to measure pressure and acceleration. In [17], an IoT-based scalable platform for storing and analyzing collected data is presented. An intelligent agricultural system that provides real-time processing of acquired big data from IoT devices is presented in [18]. The research in [19] proposed a system of cameras to record a group of lactating dairy cows, aiming to perform an analysis of cattle herd activity and to achieve working instance segmentation.

IoT identification techniques such as Radio Frequency Identification (RFID) or Quick Response (QR) codes can be applied to recognize animals [20,21] to provide more personalized animal care and more high production efficiency. The identifications have an electronic number that is distinctive to a particular organism and links the tagged animal to the relevant database [22]. In [23,24], QR code reading, processing, and display of the details are proposed. The research points out that, in this way, monitoring of several physiological parameters of an individual organism can be achieved using the collected data.

In [25]’s system, an RFID intelligent inspection terminal was developed, in which the Particle Swarm Optimization (PSO) algorithm was adopted to optimize an Artificial Neural Network (ANN) to process the monitoring data of the system by Gaussian filtering. The received classification model is based on data from RFID positioning and wireless monitoring technology. It is necessary to make full use of the data produced in smart farms to extract the full potential from the implemented high-tech systems with sensors, IoT devices, and communication modules. Although data-driven solutions have provided various benefits in agriculture, the processes of data integration, processing, and data utilization are still major challenges [4].

It can be concluded from the referenced articles that there are serious efforts being made to collect and analyze data related to smart farming. Different techniques and approaches have been chosen to perform analysis against the collected data. In addition, the researched systems are not open sourced and do not allow integration of external IoT devices or external data sources.

Visualizations play a major role when extracting useful insights from the collected data and the analyses performed on them. Systems with data visualizations built into their solutions are discussed in [26,27,28,29]. Research shows that data visualization may aid in the comprehension of large amounts of information, moreover, it is important for evaluating farming development. There are many data visualization tools available that offer many enhanced features and can work with real-time data. However, it is essential to choose a tool that suits the needed requirements [30]. A study of various chart types available for data visualization and analysis rules for the IoT domain together with an overview of the major challenges in IoT visualizations is proposed in [31].

Several publications show that a great deal of effort has been devoted to forming software architectures relating to smart farming [32]. It is necessary to note that there is no universal solution that considers the requirements in various stages of the data processing and visualization of IoT data in smart farming. In this study, it is proposed to develop a technological bridge through which farmers can easily, affordably, and comprehensibly monitor and track the development of their farms. To achieve the goal, several challenges [33,34] must be considered:Lack of standards that complicates data processing. It is difficult for one system to be compatible with different IoT devices.Data and information management. IoT devices generate fragmented, inconsistent, noisy, and heterogeneous data that must be transformed.Organizational inability to manage the complexities of IoT. Data need to be organized and visualized in an understandable format, regardless of the lack of standard analysis approaches and proper visualization of overall resultsPrivacy and security. There are four classes of cyber-attacks—data attacks, networking and equipment attacks, supply chain, and other relevant attacks in a smart farming ecosystem. Therefore, all entry and exit points of the system must be doubly protected.

The main aim of this work is to design and propose a data-driven intelligent monitoring system that can allow the integration of existing IoT devices into already built systems implemented in smart farms. The proposed architecture helps with the development of a comprehensive solution for the processes of extracting data from IoT devices, data storage, data preparation and transformation, modelling, and visualization. Streaming data from IoT devices to interactive live reports along with built-in machine learning (ML) models are included. As a result of the proposed intelligent monitoring system, the collected data and ML modelling results are interactively visualized using a powerful dynamic dashboard, which allows users to monitor various parameters across the farm and provides an accessible way to spot trends, deviations, and patterns in the data and ML results. The data-driven visualization with dynamic filters and QR code-based interactivity enables farmers to examine, organize, and represent a farm’s collected data with the goal of better serving their needs.

The proposed solution is completely cloud-based, which also helps in reducing the environmental impact of IT use [35]. Microsoft Azure is used in the design and implementation process of the proposed system. Azure is a cloud computing platform with a vast ecosystem of managed cloud services and resources such as VPC, storage, databases, networking, software, analytics, and intelligence.

According to the Gartner report [36], Microsoft is the market leader with Azure IoT, a set of services delivered on the Azure platform, among 18 others considered cloud providers. The main strengths of the platform are flexible IoT, AI/ML solution deployment approach, strong security approach, global partner ecosystem, and more. The proposed system is illustrated in the example of a livestock farm, but the developed approach can be used almost entirely in other areas of smart agriculture.

The rest of this paper is structured as follows. Section 2 introduces the materials and methods used for the architecture design for the needs of the proposed system. IoT tools are explained. An overview of the monitoring system is provided together with the included data-driven architecture concept. The QR codes as an interactivity tool to access personalized information are described. In Section 3, data-driven pipelines are explained, built, and tested. Additionally, machine learning workflow and data visualization techniques are presented. System limitations are discussed. Section 4 gives a detailed discussion of the proposed system’s elements. Finally, Section 5 concludes the article.

## 2. Materials and Methods

This study considers a livestock smart farm for the research and testing of the developed monitoring system. Most large, medium, and even small livestock farms have built-in systems that monitor certain parameters such as the amount of milk, temperature, humidity, gas level, etc., but the data are presented only in tabular raw form. They need to be stored, processed, modelled, and visualized in a suitable and easy comprehension form for any end-users, such as farmers, vets, or animal traders.

### 2.1. Materials

To achieve the objectives of this article, we used our IoT tools, that are presented in [37], and devices from two other farms with different already built and deployed IoT machines (Figure 1 and Figure 2) that collect and store their data in a database; Excel tables are used for visualizations. These devices were connected to the system presented in this paper for testing purposes.

Animal ear tags in the EU have mandatory attributes such as abbreviations of the country name and the area name. They also must include the animal registration number from the country’s local animal register. The ear tags must contain the encoded information either as a linear (1D) or initialism (2D matrix) for quick response. Tags usually have a yellow base color, and the mandatory tag attributes are lasered in black. There are no requirements for attribute positions and font size in the tag area.

Figure 3a shows various sized quick response (QR) codes in centimeters. A QR code is a 2D matrix code that is designed by keeping two points under consideration. It provides fast scanning, high data storage capacity (4296 alphanumeric characters), error correction (demanded code can also be read successfully), and many others [38]. Figure 3b shows a standard ear tag. It provides information only for the animal registration number (the same one written on the ear tag) in a text format when the QR code is scanned.

Field tests must be performed to determine the appropriate size of the QR code lasered on the ear tag. The test plan must include the lighting in the barn, contamination of the ear tag, animal’s head movements, and QR code minimum scanning distance performed with various devices such as phones, tablets, and others.

### 2.2. Methods

A concept of a cloud data-driven intelligent monitoring system for interactive smart farming is shown in Figure 4. The system allows the collection of various animal and farm data through IoT devices with sensors. Two types of monitoring, namely, remote and on-premises, are allowed. Remote monitoring makes parameter tracking available through any kind of device from different locations, and, with the help of an internet connection, the sensor data are sent to the cloud where the data are stored, filtered, processed, modelled, and visualized.

Visual interactivity is achieved using reports which show either historical or real-time data, but not both. They allow the farmer to follow the animals’ conditions and farm environmental status from the remotely monitored animals. Furthermore, the farmer is allowed to filter animals according to certain criteria and receive warning alarms when events have occurred or are expected to happen, which greatly eases the processes of animal husbandry. Both historical data (health records) and real-time data are presented only in the visualizations included in the Power BI dashboard.

Reports and dashboards can be accessed via a web interface on a computer or tablet or via a mobile app. On-premises monitoring is performed by scanning a QR ear tag, presented in Figure 5, which triggers the dynamic animal’s filter.

Personalized information for each livestock animal can be dynamically accessed instantly, on the go, through this interactive identification feature. The functionality also helps to quickly inform the veterinarian or farmer about the animal’s current and previous health status.

### 2.3. Data-Driven System Architecture

Intelligent systems include a flow of behavior between hardware and software components [39]. The design of intelligent systems includes some key characteristics [40,41]:-Sensors—collect data from the environment and transmit it to the system core for identification and analytics.-Identification—intelligent systems must automatically recognize specific information and transmit it to subscribed services or channels.-Data analytics—an essential function of an intelligent system is its ability to process collected data.-Self-learning—the intelligent system needs to include artificial intelligence or machine learning functionalities.-Real-time communication—an intelligent system needs to have the ability to simulate or emulate in real-time or near real-time.-User Experience (UX)–to interact with users, intelligent systems must have interfaces such as web pages, reports, dashboards, or other types of visualization.-Remote system management—an intelligent system allows users to interact with it from any location.-Interoperability and connectivity—an intelligent system must combine its elements into a holistic communication process.-Protection—an intelligent system’s ecosystem, networks, and communications must be secure to be available and reliable to function properly.

Figure 6 shows the developed architecture of the proposed intelligent monitoring system which includes the key characteristics for the design of an intelligent system. It consists of an IoT part, various cloud services, and the connections between them. Implementing the entire process from data extraction through storage, filtering, processing, forecasting, and visualization is very complex and requires a thorough selection of cloud services and the methods of communication between them.

In the development process of the built cloud architecture, the five pillars of the well-architected systems [42] were followed. They are optimized for reliability, security, cost optimization, operational excellence, and performance efficiency. They describe the design principles and the best architectural practices for designing and executing cloud workloads. Following these pillars helps produce a high-quality, stable, reliable, scalable, and efficient cloud system.

The architecture consists of two parts—one built on-premises with IoT devices and the second deployed in a cloud environment. The Cloud part is developed by using six interconnected services. All services are selected to be serverless because they need no server management and are inherently scalable, quick to update, feature decreasing latency, and lead to significant cost reduction. Each service has a clear bounded context and single responsibility, and each service may work with the result of the others. The services used [43,44,45,46,47] in the presented architecture are as follows:-IoT Hub—enables a reliable and secure bi-directional communication process between IoT devices and the Cloud. It is a working IoT service which is hosted in the Cloud. In addition, the IoT hub provides scalable device-to-cloud and cloud-to-device messaging and secure communications. The IoT Hub is the only external entry point for the system. Part of the IoT Hub is the Defender for IoT service, which provides device registration and identification along with comprehensive monitoring and alerts in case of threat detection against IoT environments. It is a standalone service designed to add an additional layer of threat protection for the IoT Hub and registered IoT devices. In addition, IoT Defender uses masked IP addresses and processes and stores the data in a different geographical location from the IoT Hub. The IoT Hub is also responsible for synchronizing the received data from the IoT devices and propagating them forward to the subscribed services.-Stream Analytics—Stream Analytics is a stream processing engine that processes large volumes of fast streaming data with minimal latencies from multiple sources simultaneously. It can work with data from a variety of input sources including IoT devices, driverless vehicles, and different applications. The output can deliver data to another service for alerts and actions, dynamic dashboarding, data warehousing, and storage. Stream Analytics is the logical part of the architecture. All data that pass through it are filtered based on predefined queries, sorted, and then transmitted to the subscribed services.-Blob (binary large object) Storage is object data storage. It is part of Azure DataLake. In addition, it is optimized for storing massive amounts of unstructured data, such as text, images, IoT data, video, and audio. There are two types of blob storage—hot storage and cold storage. The hot tier of storage is for storing frequently accessed data—data that require high durability, availability, and quick time to access. The cold storage tier is for storing data that are long-lived and infrequently accessed. Blob storages are used for persisting raw and filtered data and system operational logs.-The machine learning service is completely cloud-based and self-contained. It features a complete development, testing, and production environment for quickly creating predictive analytic solutions. It is the data analytics and machine learning part of the system. In it, the data are transformed and modelled to build a machine learning model to predict future values.-Power BI is an interactive data visualization and business intelligence tool that converts data from different data sources to interactive dashboards and dynamic reports. Data can be entered by reading directly from a database, streamed by IoT devices, webpage, or structured files such as excel spreadsheets, CSV (comma-separated values), XML (Extensible Markup Language), and JSON (JavaScript Object Notation). Power BI is the representation part of the architecture. Interactive reports and dashboards are designed and built. They visualize machine learning predictions, real-time data, and statistical information based on raw historical data using charts, tables, and other visual tools.

## 3. Results

The proposed data-driven monitoring system shown in Figure 6 is cloud-based. It depends on Azure Cloud services and technologies and infrastructures such as Azure IoT Hub, Blob Storages, Streaming services, Azure Machine Learning Studio, Power BI, and others. Every service used has its own responsibility, following the single responsibility approach. Using services and technologies from a single provider also helps to achieve consistency between the development, deployment, and maintenance of the proposed system.

The system is built using the following steps:Building data-driven pipelines in Azure Cloud:○Setting up Azure IoT Hub to accept the registration of IoT Devices.○Setting up Azure Storage Account with Blob Storages to persist IoT historical and real-time data.○Setting up Azure Stream Analytic to process IoT data.○Monitoring and testing the data-driven pipelines.Design and implement the Azure Machine Learning Workflow.Design and build interactive Power Bi reports and dashboards.Integrate QR codes with Power Bi interactive reports.Define system limitations.

### 3.1. Data-Driven Pipelines

The data-driven pipelines are realized using Azure Cloud services such as Azure IoT Hub, Stream Analytics, and Blob Storages.

#### 3.1.1. IoT Hub for IoT Devices Registration and Communication

This is a service which allows data exchange between physical IoT devices and Cloud based services. It is fully managed by Azure, but it requires a corresponding setup according to user needs to allow for IoT device provisioning.

Provision of IoT devices to the Smart livestock monitoring system is achieved by registering the devices to the IoT Hub. The IoT Hub identity registry is used to create device identities and credentials which provide unique device identification.

IoT devices connected to the IoT Hub are managed using built-in functions such as:-IoT device metadata and state information for all connected devices are stored, synchronized, and queried.-IoT device state is set either per device or in a group depending on the farm structure.-State change in IoT devices can be automatically responded to by using message routing integration.

For each IoT device registered with the IoT Hub, a unique device ID, primary key, and connection string are automatically generated. The unique connection string is based on the primary key, and it is used for device authentication in all API calls during the device communication with the IoT Hub. The successful connection of an IoT device must follow two main steps. Firstly, all necessary Azure IoT packages need to be installed or referenced in the IoT device and, secondly, the generated primary connection string must be set in the code to allow authorized communication against the IoT Hub.

The IoT hub has the possibility to communicate outside the cloud environment. It uses IoT Defender for additional protection to provide end-to-end threat detection for IoT/OT environments.

#### 3.1.2. Azure Storage Account with Blob Storages

Azure blob storages are cloud solutions to store various types of unstructured data [48]. It is associated with an Azure Storage Account which provides tools to uniquely identify stored objects in the Azure infrastructure.

The proposed system needs two blob storages—one to store raw data, the other one to store prepared and filtered data. The blob storage that stores raw data is connected to the Power BI service for creating dynamic reports with information from the present and the past. The blob storage that stores filtered and prepared data is connected to the Azure Machine learning studio.

#### 3.1.3. Stream Analytics

A scalable real-time analytics service is provided by Azure. Jobs are used to build streaming pipelines to ingest data from various data sources and persist them in native cloud storages. It also has built-in machine learning capabilities [49].

The jobs are created in the Stream Analytics service, which consists of an input, query, and output. The job uses ingested data from the IoT Hub as an input. The query is based on the SQL query language for sorting and joining queries on streaming data. The stream job has several outputs such as blob storage and Power BI service.

Figure 7 shows a data pipeline from the IoT Hub used as input through Stream Analytic Job (query) to Power BI (output). The metrics use a 30 min time window. There are 123 registered input events in the IoT Hub for that window. The sum in bytes for all input events is 30 kBs. These events are ingested by the Stream Analytics Job service where the query is executed. There are no filters or data sorting set up in the current query, and the entire set of events is transmitted to the output service (Power BI). As a result, the Power BI service received 122 of 123 events. The last event is still in the process of receiving due to a 6 s delay (the delay period is variable).

Figure 8 shows the data pipeline from the IoT Hub (input) through the Stream Analytic Job (query) to Blob storage in a Storage Account (output). In the 30 min time window, there are 122 input events registered in the IoT Hub. The sum of input event bytes is 29.8 kBs. Backlogged input events and input deserialization errors are 0. This process is performed twice. In the first process, the query in the Stream Analytics Job omits all data from being directly written to blob storage. In the second process, the query contains a selected set of variables and has an additional “where” clause set to remove null and invalid values. As a result of the processes, the storage account received 122 out of 122 events. The watermark delay is 0 s. There are two blob containers created in the IoT resource group space. Each of them includes “logs” files and “streaming data” files with a hot access tier in CSV format.

#### 3.1.4. Monitoring and Testing of Data-Driven System Pipelines

Monitoring of the created pipelines is needed to detect such conditions as input events per second, output events per second, late events per second, number of runtime errors, and more. The metrics are used for setting up initial benchmarks for the data pipelines testing processes.

Jobs can be difficult to monitor because some streaming jobs may have complex and unpredictable patterns of incoming data and the output they produce.

The proposed data-driven system architecture involves data from IoT devices that send data periodically (scheduled process), therefore, the IoT Hub receives data only during specific intervals and produces an output when new data exist or some rare condition occurs. This process provides prerequisites for the occurrence of errors in the transmission of data between services—especially in a streaming analytics service. Such errors can be easily spotted in the system metrics available for the pipeline.

The important metrics showing the data processing performance and success rate are:-The sum of input and output events that are deserialized and sent to the target,-The delay measurement between input and output events in the stream job,-The number and type of errors that occur in the stream job,-The percentage of processor utilization and memory utilization during job executions.

Some of the most common types of errors are runtime errors, input deserialization errors, and data conversion errors. The runtime errors are errors related to query processing. They exclude errors found while ingesting events or outputting results. The input deserialization errors are input events that could not be deserialized. The data conversion errors are types of output errors where output events could not be converted to the expected output schema. Due to the error policy, they can be changed to “Drop” events.

Figure 9 shows the sum of input and output daily events in the Stream Analytics job. The input events are the number of records deserialized from the IoT Hub. The output events are the amount of data sent by the Stream Analytics job to the output target in a number of events.

As the figure shows, the total number of output events is 852 out of 852. This indicates that the job is performing correctly and there are no losses. The entire volume of data passes correctly to the output destinations—Power BI and Storage account.

Another major challenge is the watermark delay that indicates the delay of the streaming data processing jobs. The watermark delay is across all partitions of all outputs in the job. This is of great importance for any system. Figure 10 shows minimum, maximum, and average values for the watermark delay. The minimum delay is 0 s, the maximum is 7 s, and the average delay is 5.19 s. The delay between the current time and watermark delay is relatively small, which is a clear indicator that the job is keeping up with the incoming data and producing the query results on time.

In case the watermark delay goes over the set-off threshold, which is 40 s, the number of SU (system memory utilization) needs to be increased or the query needs to be parallelized.

Figure 11 shows that during the streaming job, there are zero errors (runtime, deserialization, data-conversion). These results are prone to change, and they need to be monitored permanently. Monitoring and a timely interception, tracking, and correction of errors are some of the most essential actions in an improved type of process. The chart shows the lack of errors for a short period of time. However, the chart can be useful for easily spotting errors in long-term series if they occur somewhere along the pipeline. In general, infrastructure-related errors are unlikely to be expected as cloud providers are well-known and ensure that they provide services with high levels of durability, reliability, and availability. Data conversion errors can occur when messages received into the system are incorrectly formatted. The structure of the message is controlled by the logic implemented in the IoT device. All messages have a uniform interface and are sent to the system as JSON objects.

Additional important indicators of the correct operation of data pipelines are CPU% Utilization (central processing unit) and SU % Utilization (system memory utilization). Figure 12 shows the percentage of CPU utilized and the percentage of memory utilized by the created stream job; they are, respectively, 78% and 8%. These two indicators are related to each other. For effective work, the CPU percentage should not exceed 90%, and the SU percentage should not exceed 20%. These metrics have spikes intermittently, which is acceptable.

### 3.2. Azure Machine Learning

Following the system architecture presented in Figure 6 in Section 2, the stored selected data in blob storage are imported into the Azure Machine learning service. Before developing a machine learning model for predicting the future quantity of milk, the imported data are cleaned, normalized, regularized, trained, and tested. The whole workflow is presented in Figure 13. The presented workflow is the way to get work done, and is illustrated as a series of steps that need to be completed sequentially.

The farm data used to guide machine learning model training are collected through various IoT devices and tools. Data manually entered by farmers into Excel spreadsheets are also used.

The data consist of observations for a period of over 2 years. They have, in total, 2,285,665 values organized in 43 columns and 53,155 rows. Data are in a raw format and have several data types, such as datetime64[ns] (4 columns), float64 (13 columns), int64 (7 columns), and object (19 columns). Of the values, 11% are null values.

For the correct implementation of this workflow, it is necessary, on the one hand, to correctly choose the work features (columns) from the input and the predicted variable (output) in step “Select columns in database”. “Daily” is the target variable for prediction representing a daily quantity of milk per cow. Statistical characteristics are as follows: mean—33.115; standard deviation—13.113; minimum—0.00; 25%—24.600; 50%—33.700; 75%—42.400; max—77.400; 70% non-null values.

On the other hand, it is necessary to make a correct preliminary selection of the used algorithm. Experiments to select suitable methods and ML algorithm must be conducted separately in another testing workspace.

All text-based features are converted to category type and, subsequently, to int64, and each category is assigned a unique value. The seasonality of the data is also considered.

Based on some features, new features have been created to accommodate the proper training of the machine learning model.

The Multivariate Imputation by Chained Equations (MICE) method is used to perform the next step, “Clean missing data”, where a new value is assigned for each missing value, which is calculated by the MICE method [50], in which a condition is modelled separately for each missing data variable, using data from other adjacent variables before to fill in the missing values. MICE are considered due to data heterogeneity and seasonality. Using other available methods such as mean replacement, median, mode, and others introduces noise or significantly reduces the total number of observations used.

The normalization of the data is performed at the next stage. Z-score (1) is the chosen method for data normalization because it handles outliers better. Unlike other methods such as Standard Scaler and Min–Max, the Z-score indicates how many standard deviation units an individual observation is away from the mean. It performs calculations in confidence interval (−3*σ*: 3*σ*) [51] (p. 64).
(1)$$Z=\frac{x-\mu }{\sigma }$$
where *x*—raw feature value, *μ*—mean value of the feature, and *σ*—standard deviation of the feature.

The fact that *z*-scores belong to the standard normal distribution makes it possible to use z-scores to compare heterogeneous values of primary measurements.

The cleaned and normalized data are fed to the next step, “Filter Based Feature Selection”, which uses different statistical calculations to determine a subset of features with the highest predictive power. The selected feature scoring method is Pearson correlation (2) where, for every two variables, a value is returned that indicates the strength of the correlation. This method was chosen over its alternatives such as the Spearman and Kendal correlation because the data were normally distributed and due to the prediction of numerical but not categorical results. Furthermore, this method is not affected by changes in the scale of the variables used [51] (p. 66).
(2)$$r=\frac{n\sum^{\;}xy-(\sum^{\;}x)(\sum^{\;}y)}{\sqrt{[n\sum^{\;}x^{2}-{\left(\sum^{\;}x\right)}^{2}][n\sum^{\;}y^{2}-\left(\sum^{\;}y{)}^{2}\right]}}\;$$
where *n*—number of pairs of scores, ∑*xy*—sum of the products of paired scores, ∑*x*—sum of *x* scores, ∑*y*—sum of y scores, ∑*x*^2^—sum of squared *x* scores, and ∑*y*^2^—sum of squared *y* scores.

As a result of this step, twelve features were outlined that had the strongest predictive power and could be used to predict the amount of milk produced. These are Lactation (the period between one calving and the next), Age, DIM (days in milk), Dry (a stage of their lactation cycle where milk production ceases prior to calving), Season, Humidity, Temperature, Breed, Animal Noise Level, Motion, Days in Group, and BCS (Body Condition Scoring).

The ready-transformed data are trained and tested with the Boosted Decision Tree algorithm. Boosted Decision Tree Regression is a ML algorithm for solving regression tasks that creates a prediction model in the form of an ensemble of decision trees [52,53], thereby allowing prediction of the value of the target variable by learning simple decision rules, inferred from data features. A decision tree is a structure in which each internal node represents a “test” of an attribute; each branch represents the result of a test. It is suitable for datasets when there are multiple training features.

The Boosted Decision Tree algorithm builds the model in stages, as other boosting methods do, and generalizes them, allowing optimization of an arbitrary function of varying loss. Each tree is created iteratively:○The output of the tree *h_t_*(*x*) is given a weight *w* relative to its accuracy,○The output of the ensemble is the weighted sum:
(3)$$\hat{y}\left(x\right)=\sum_{t}w_{t}h_{t}\left(x\right)$$○After each iteration, each data sample is given a weight based on its classification. The goal is to minimize the function:
(4)$$O\left(x\right)=\sum_{i}l\left({\hat{y}}_{i},\;y_{i}\right)+\sum_{t}\Omega \left(f_{t}\right)$$ where $\;l\left({\hat{y}}_{i},\;y_{i}\right)$ is a loss function—the distance between the truth and the prediction of the *i*-th sample, and $\Omega \left(f_{t}\right)$ is a regularization function—it is responsible for the complexity of the *t*-th tree.

The model is trained and tested in the “Cross-validation model” where the dataset is divided into 10 subsets (folds) [51] (p. 88). The model is built on each data subset, then, a set of statistics is returned to estimate the performance of each fold. The whole dataset of data for training and testing is used. The resulting coefficient of determination (R^2^) is 0.897885, which is used to evaluate the performance of the model.

The ready-to-use model is then deployed in Azure Machine Learning as a web service. The model is accessible in Power BI as a web source and loaded in Power Query Editor [44] (pp. 77–81).

There are some events that can trigger model updates. The first one is triggered if the performance of the model starts to deteriorate at some pre-determined threshold. The process of retraining and fine-tuning the model is then triggered. Another is triggered if the data schema changes and new features are introduced into the model.

The model is retrained with the new data and the historical data before the model is deployed into production. The difference is that the already known model parameters are used as a starting point during the retraining process. This significantly reduces the time to create a production-ready machine learning model.

### 3.3. Power BI Data Visualisation (Reporting and Dashboards)

Power BI is the interactive part of the proposed system. By designing and creating reports and dashboards that contain both raw and processed data, end-users (farmers, veterinarians, etc.) are provided with understandable visualizations. Reports and dashboards in Power BI have different purposes and allowed functionalities.

There are two types of reports. The first type contains historical data. The second type contains real-time data. Reports with historical data can accept new data (updates) according to a predefined schedule refresh period—for example, every day at certain times.

The reports presenting data in real-time (live connection) update the data automatically as soon as they are ingested by the system and are made available in the Blob Storage. Reports’ contents can be filtered by using inbuilt or custom filters. The content can be accessed via web devices or via QR code.

A dashboard can contain elements from several reports or custom tiles with predefined data. It can combine historical and real-time data. The filtering functions are not allowed. It cannot be accessed via QR code, only via web devices.

Following the designed architecture in Section 2, Power BI is the destination in the data pipeline. It receives data from three sources. The first source is the Azure Machine Learning studio, which sends a ready machine learning model. The second source is the Stream Analytics service, which pushes real-time data. The third source is Blob storage, which sends raw data. The data from the first and last sources are combined into one common report, shown in Figure 14 and Figure 15. The real-time data are shown in Figure 16 as a separate report. The combination between reports is presented in the dashboard in Figure 17.

Figure 14 shows the analytic report. It assists the user in obtaining information on the total number of animals, sex, breeds, average age of the herd, average milk yield, distributions and groups of animals, number and type of animal morbidity, and a number of pregnant animals and others.

By activating any report filter, complex DAX (data analysis expression) queries are executed to filter data for a specific animal from a farm, and the user can access information about the age, breed, minimum, maximum, and average amount of milk, types, and frequency of diseases (events), number of pregnancies (lactation), and more. Based on these data, various statistical parameters are calculated, which support the process of assessing the health status of the animal. The analytic report updates the data twice daily. The data are fetched from the Blob storage.

Figure 15 shows a visualization of the developed regression machine learning model in the Azure Machine learning studio. It is visualized by a linear diagram. The *x*-axis shows the data time, and the *y*-axis shows the daily quantity of milk. The set forecast period is 15 days. Results presented over a longer period would be inaccurate and may be misleading. This is because of the specificity of the predictor variable.

In order to improve and maintain the developed model, it is important to also work with new data, and for that, the data’s scheduled refresh period in Power Bi is set to occur on a daily base. When the new data are fetched, Power Bi also triggers the Azure Machine Learning endpoint to evaluate and score against the new data.

Figure 16 shows a real-time report with data coming from livestock IoT devices. The report receives data directly from the stream analytics service, with a minimal delay following FIFO (First-In, First-Out) methods [54]. The report’s visualizations are updated as soon as the IoT devices are turned on and start sending data to the system. The ingested data are stored in a temporary table of 200,000 rows due to the Power Bi retention policy.

In addition, notifications are included in the real-time report. They are activated in any of the following cases, specifically, data are detected that exceed or fall below the pre-set allowable low/high measurement values, or there is an error updating the data and the reports are updated.

Figure 17 shows a dynamic dashboard which contains tiles from the real-time report and tiles from the historical data report. The purpose of the dashboard is to present all the important information in one place to make the monitoring process easier, faster, and easier to comprehend. The dashboard is the only way to combine, in one place, different types of information (real-time data, calculated data, raw data).

Since the dashboard is designed and built with the purpose of presenting all the data in a single place, the F-pattern design [55] is followed. As the human user does not read the information but scans it, following this design pattern, it is easier for the user to identify the significant information.

### 3.4. QR Code

The presented system offers new functionality to track the animal’s vital data by scanning a QR code associated with a filter to select and open a report with personalized dynamic information about the animal (health card). This functionality helps to quickly inform the veterinarian or farmer about the current and previous health conditions of the animal (for which data have been collected in the system).

Figure 18a shows testing of the customized QR code. The testing results showed that a QR tag with dimensions height = 2 cm and weight = 1.5 cm has a maximum scanning distance by plain phone camera of 50 cm and 80 cm by tablet. The maximum scanning distance increases to 1.80 m with a phone with a special application for reading QR codes and 2.10 for the tablet. The illumination level of the farm is constant, and shadowing is not observed. Figure 18b shows QR code testing performed with two types of devices—tablet and phone. Testing results showed that the custom reports opened correctly on both devices. Contamination of the ear tags interfering with the scanning process was not observed. The movement of the animal’s head is smooth and also did not obstruct the scanning process.

Based on the test results, it is confirmed that the newly built functionality of the system can be applied in real conditions, which makes the use of QR codes on ear tags worthwhile.

Due to the preference of farmers to stick to only one breed of animal on farms, animals can be very difficult to distinguish from one another. This makes the individual monitoring process slow and complicated. By using QR tags, this process is much easier and, thus, preferred by many users.

### 3.5. System Limitations

The system is currently configured for a maximum of 400,000 messages per day per IoT device. Currently, the maximum number of devices that can be connected to 1 IoT Hub is 200. The maximum limit for IoT Hubs is 50 per standard Azure account. Therefore, the system limit is 10,000 IoT devices. The total number of daily messages is 4 × 10^9^. The maximum size of a single device-to-cloud message is 256 KB. The maximum size of a single cloud-to-device message is 64 KB.

## 4. Discussion

### 4.1. Materials and Methods

The concept of an intelligent system was explained and visualized (Figure 4). Following the concept, the functional requirements of the system were outlined. An architecture design for an intelligent monitoring system was developed. Before the architecture design, research was conducted on the preliminary analysis of smart farming problems, other similar systems, and available cloud providers with available services. Azure was chosen as a cloud provider. Six Azure services were selected from over two hundred others. The genuine authors’ approach was used for services’ organization, setup, communication methods, and roles in the system. The architecture was implemented during the development of an intelligent livestock monitoring system. It can also be applied during the implementation of other systems integrated into smart farming.

### 4.2. Results

The proposed architecture was implemented by building automated Azure data-driven pipelines, presented in Figure 7 and Figure 8, and the following services—IoT Hub, Stream Analytics, Machine Learning, Blob Storage, and Power BI. Queries were implemented in the logic block of each pipeline. The logic block controls data flow based on the system requirements. Every pipeline was deployed in a production environment.

#### 4.2.1. Azure IoT Hubs

Fleets of IoT devices help humanity to live in an interconnected world, where major industries such as transportation, automation, medical, agriculture, and others cannot allow failures. The built Azure Cloud-based IoT Hub is the service which greatly reduces the maintenance burden of IoT fleet management by providing centralized capabilities for registering, operating, monitoring, and updating a large number of IoT devices. Centralized IoT device management solutions also allow immediate insight to be gained on overall IoT systems’ performance and reduce security risk. The IoT Hub has an identification feature here, which is one of the main key features in developing an intelligent system.

#### 4.2.2. Stream Analytics

Real-time data processing and communication is the key technology to achieving real-time actions and interactivity. Furthermore, it is a key characteristic of an intelligent system. This is accomplished with the proper stream analytics suite. It needs to be a cloud-based one, in the context of the millions of interconnected IoT devices and the tons of messages those devices generate and push forward for processing and analytics.

The built Stream Analytics service in the system is responsible for receiving the data from the IoT hub and their logical distribution. It is an essential part of any pipeline built into the system. There are three endpoints’ services to which it transports the data—raw data storage, data visualization, or data processing and modelling. In addition, the article presents the results of the monitoring process of the constructed pipelines (Figure 7 and Figure 8). They show that for the test period, the pipelines operate without any noticeable errors, with a normal load and an average delay of 5.19 s. In summary, the built service is scalable, reliable, and time sensitive.

#### 4.2.3. Azure Machine Learning

The Azure Machine Learning Studio was the selected tool for developing ML models. It was provided by Azure as a software as a service (SaaS). This is an important feature because Azure takes care of provisioning and maintaining the computational, storage, and global infrastructure. It contains two of the key characteristics for developing an intelligent system, which are data analysis and self-learning.

As a result of the applied approach presented in Figure 13, the ingested data were cleaned, transformed, and normalized. Data loss and null values were significantly reduced. A regression model was created to predict future milk quantities. The model was implemented as a web service. The performance of the model was monitored to detect problems with the predictions, such as overfitting and underfitting.

#### 4.2.4. Power BI

All data insights are important as they bring business value to critical decision-making. Insights must be intuitively reported and visualized. If any of the reporting and visualizations are incorrect, that will greatly impact the decision-making process in the negative way.

Power BI is the selected service which works with data storage and streaming sources and provides intelligence and insights into the created uses of interfaces to support the user’s decision-making. Automated reports have been developed which show machine learning predictions, real-time data, and statistical information based on raw historical data. The results are organized through various interactive visual elements such as charts, tables, graphs, and others. The content of all reports is dynamic. Custom filters using complex DAX queries are built into each report, which allow for filtering of data by time periods and different categories such as farm, herd, animal numbers, breed, etc.

The F-pattern design was based on the hypothesis that a human user does not read the information but instead scans it. A comparison was made of the advantages of the F-pattern design over others. A dashboard was designed and built with which the necessary information was obtained by scanning the content, but not by reading it. This makes it easier for the user to spot significant information. The design and structuring work of the visuals was performed with farmers’ satisfaction in mind. Numerous alpha–beta tests were performed on the dashboard reports and visualizations, which resulted in a greatly improved user experience.

Publish capabilities allow native usage of created reports and the dashboard on different types of devices such as laptops, desktops, and mobile. This service has two of the key characteristics of an intelligent system, which are user experience and remote system management.

New functionality was built, which allows access to specific information per animal using QR codes and a filter which is integrated into a dynamic report. Manual tests were performed on the developed QR codes to determine their appropriate size and the distance from which they can be scanned and the encoded information recognized.

#### 4.2.5. Pipelines Tests and Monitoring

Tests and monitoring of the pipelines’ performance were performed. Key characteristics such as data loss during data transmission between services (Figure 9), data transmission delay (Figure 10), errors (Figure 11), and CPU load (Figure 12) were monitored and analysed. The results showed a lack of data loss. The transmission delay had an average of 5.19 s. No errors occurred during the pipelines’ operations. System load levels were normal.

### 4.3. System Limitations and Costs

Every system has limits. Developing cloud-based IoT systems has the benefits of knowing the limitations upfront and designing better solutions according to the known limitations. On the one hand, cloud scalability greatly improves the design process, and system limitations can be easily overcome by scaling cloud services to the desired capabilities. Of course, everything comes at the cost of more expensive cloud infrastructure. The main difference is that the cloud system can scale on demand, and system expenses can be easily predicted and planned upfront.

The main limitation of the proposed system is the number of devices for an IoT Hub and the throughput of the stream analytics jobs. Both can be scaled to the desired capabilities if needed.

Azure Cloud works according to the OpEx (Operating Expenses) model, which allows the system to scale up or down to meet the specific capacity needs. Unlike the CapEx (Capital Expenditures) model (which is used for on-premises systems), where the upfront cost of physical infrastructure and expense planning is required at the start of the project, in the OpEx model, resource usage is charged based on the usage following the “pay-as-you-go” model. One major benefit of the OpEx model is that there are no wasted resources, because if resource usage is not needed, then it is not provisioned and, subsequently, not paid.

The system uses a standard pricing tier (S1):

IoT Hub—$25 monthly per IoT Hub unit.

Defender for IoT—$0.001 monthly per connected device.

Stream Analytics—$1 monthly per device.

Blob Storage (Hot)—$0.02 monthly per GB for first 50 terabyte (TB).

Azure Machine learning—$9.99 monthly per ML studio workspace and $1 per studio experimentation hour.

Power BI Pro—$9.99 monthly per user.

### 4.4. Research on the Broader Integration of IoT Systems in Smart Agriculture

There are many IoT solutions that do not have a data analytics platform and cannot utilize the full potential of the gathered data. It is very important to popularize such cloud-based analytics platforms to help integrate more IoT systems and stand-alone devices. The more data, the better analysis is performed. IoT devices use well-known protocols and work with well-known data types and formats. Using the Cloud’s virtually limitless scalability options, data pipelines in analytics systems can easily ingest data from more IoT devices and sources, which will greatly reduce the cost of the system and help perform better analyses and achieve better overall results.

### 4.5. Privacy and Security

The use of IoT devices and intelligent communication technologies increases cybersecurity threats and vulnerabilities in smart farming environments that have specific attributes such as farm equipment, labor sharing, and operational approaches and decisions.

Collected farm-specific data such as farm location, yields, number of animals and their conditions, available quantities of products in the warehouse, equipment, staff shifts, and more are of interest to competitors, distributors, prospective investors, and others with malicious intent. Therefore, the development of intelligent farm systems requires more attention and a high level of security. The proposed system was developed in a cloud environment due to the high level of security present at every layer.

IoT defender is a supplementary service used for an additional level of security for both the IoT Hub and individual IoT devices. This way, when a physical attack is performed on an IoT device, the attacker will not receive any sensitive information, and the owner will be immediately notified.

Gaining control over insecure IoT systems also can be used later for launching large-scale DoS attacks against other internet-based systems, causing huge damages [56]. Such cyber-attacks have the potential to disrupt the economies of countries that are heavily dependent on smart agriculture.

### 4.6. Future Work

The system is planned to be improved and enriched with new functionalities, such as:-Research and design of new features for ML models.-Research and development of approaches, techniques, and tools to automatically calculate BCS (Body Condition Score), to better track the health status and fat levels of an animal along with the general condition of the herd.-Research on automatic analysis of daily animal activity to improve early detection of animal diseases.-Designing and building additional reports and dashboards for monitoring and improving their interactivity.-Automatic generation and export of daily, weekly, monthly, and annual reports with statistical information, both for the entire farm and for each individual animal. Automatic comparative analysis against generated results.

After the future development of the functionalities, it will be necessary to add new services and increase the capacity of the system.

## 5. Conclusions

The data-driven intelligent monitoring system for interactive smart farming presented in this article is pure cloud and based on the Azure Cloud environment. It uses various serverless services. The system is open to working with various external IoT devices and data sources. With this approach, the system can resolve major problems for farmers, such as utilizing the full capacity of the gathered data and IoT devices and tools which are purchased under various programs related to technological modernization in agriculture.

The proposed architecture design has a strict data flow with well-defined data consumers. The system has only one well-defended external entry point, which significantly increases the system’s level of security. In the intelligent part of the system, a machine learning model has been developed that predicts future milk quantities. The interactive reports and dashboard created for the user allow remote and on-premises monitoring. Using intuitive and understandable visual elements in the created reports and dashboard, both historical data and real-time data along with machine learning predictions are presented. The system can control the IoT devices to send data on demand and set alarms to send notifications when certain events occur. New functionality is proposed which allows access to specific information per animal by scanning a QR code and provides full-fledged use of QR codes located on the ear tags. The solution is in support of the green environment. The Azure Cloud offers green computing and carbon footprint lowering options. The remote servers are located in places with cooler climates and natural availability of water. In this way, servers’ cooling costs can be significantly decreased, which greatly reduces the environmental impact.

The proposed system is a keystone in the technological bridge connecting farmers and modern technological data-driven solutions.

## Figures and Tables

**Figure 1 sensors-22-06566-f001:**
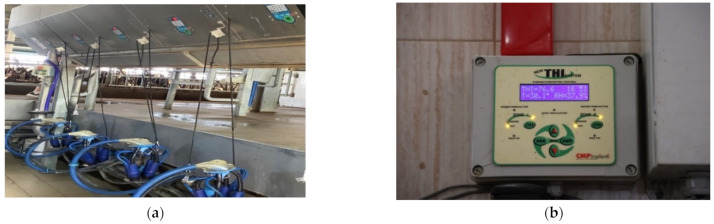
(**a**) An automated milking system that records the daily amount of milk for each cow; (**b**) a farm environment management system. It measures temperature (T), humidity (RH), and temperature humidity index (THI).

**Figure 2 sensors-22-06566-f002:**
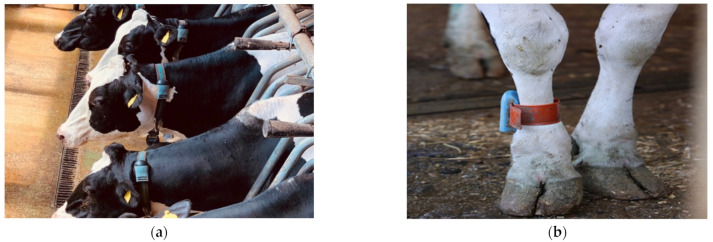
(**a**) Cow neck device with sensors for movement and (**b**) cow leg device with sensors for heat detection, animal identification, and calving alert.

**Figure 3 sensors-22-06566-f003:**
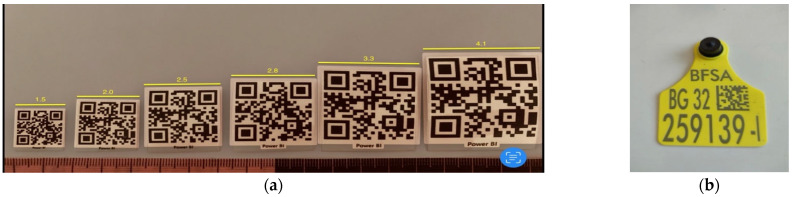
(**a**) QR codes with different sizes in centimeters, and (**b**) standard ear tag with QR code.

**Figure 4 sensors-22-06566-f004:**
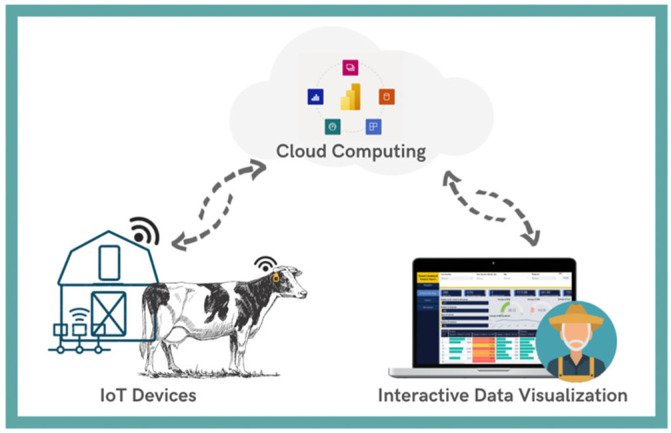
Smart Livestock Monitoring System concept.

**Figure 5 sensors-22-06566-f005:**
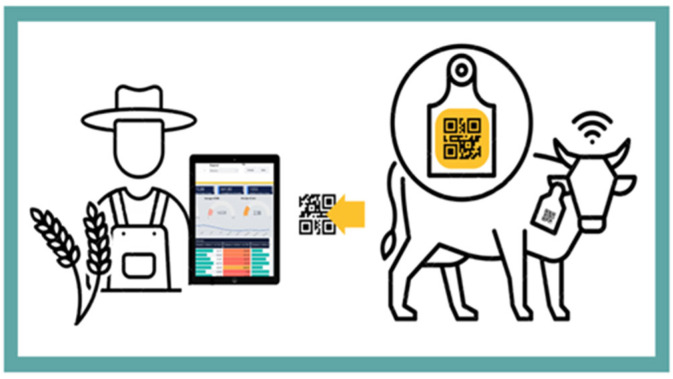
Personal identification by QR code.

**Figure 6 sensors-22-06566-f006:**
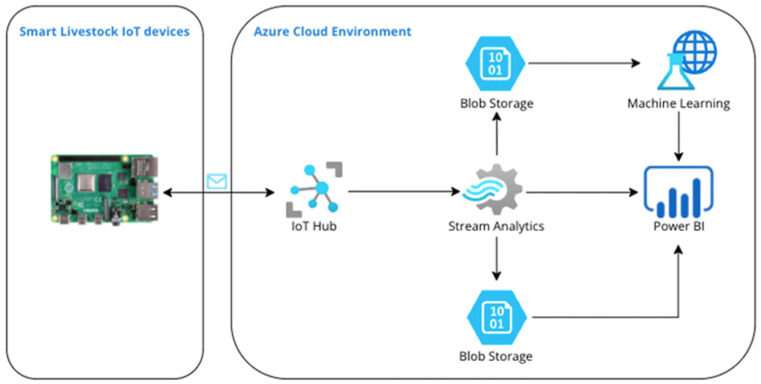
Smart Livestock system architecture.

**Figure 7 sensors-22-06566-f007:**
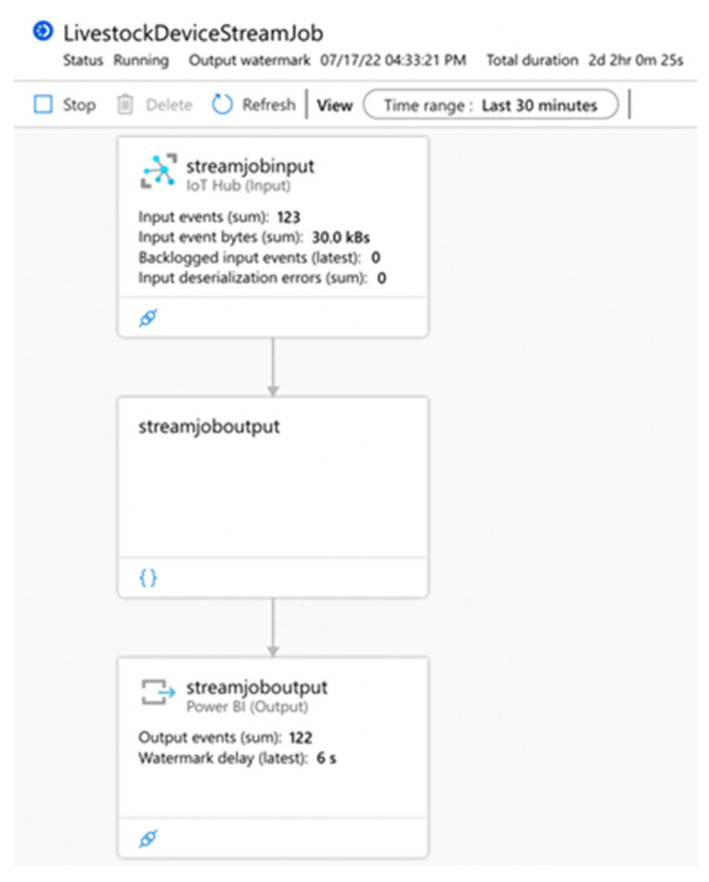
Data pipeline—IoT Hub (input), Stream Analytic Job (query), Power BI (output).

**Figure 8 sensors-22-06566-f008:**
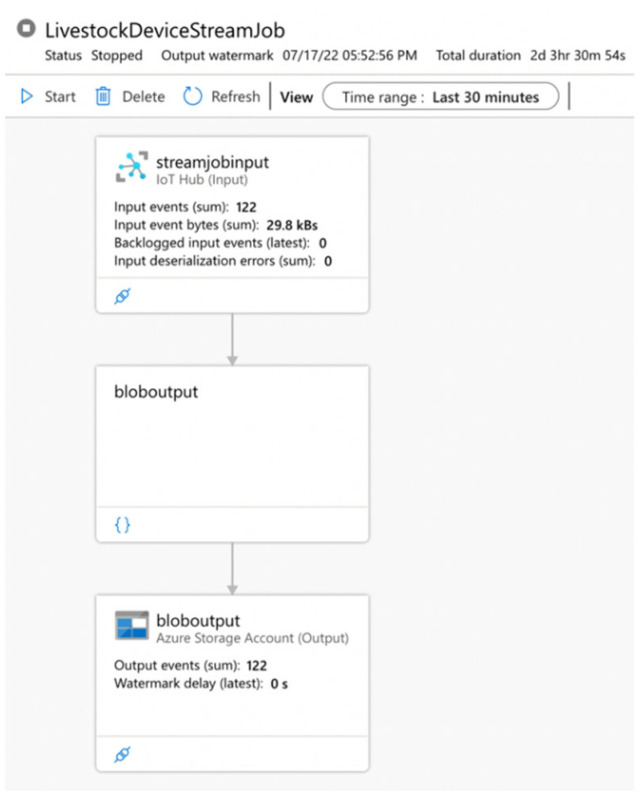
Data pipeline—from IoT Hub through Stream Analytic Job to Blob Storage.

**Figure 9 sensors-22-06566-f009:**
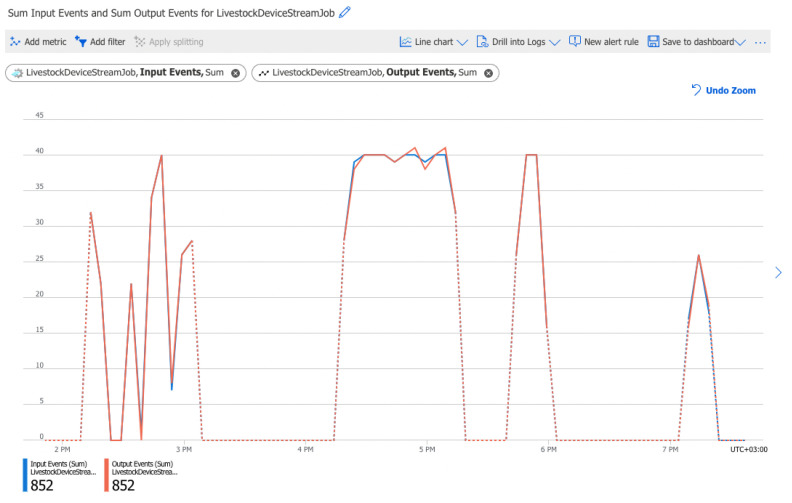
Sum input events and sum output events for livestock device stream job per day. The blue line shows the sum of input events. The Red line shows the sum of output events. The dotted line shows that no data is ingested for that period (when IoT devices are in sleep mode).

**Figure 10 sensors-22-06566-f010:**
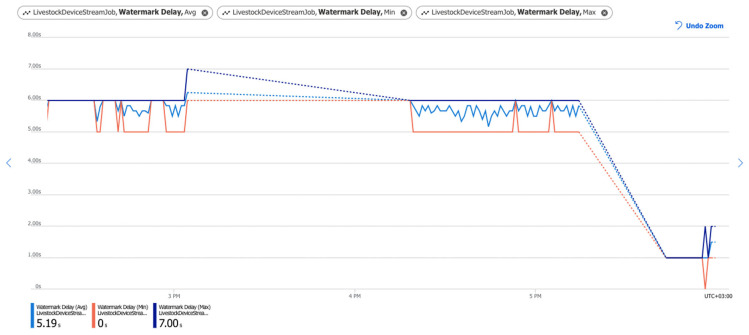
Watermark delay. The blue line shows the average watermark delay. The Red line shows the minimum watermark delay. The dark blue line shows maximum watermark delay. The dotted line shows that no data is ingested for that period (when IoT devices are in sleep mode).

**Figure 11 sensors-22-06566-f011:**
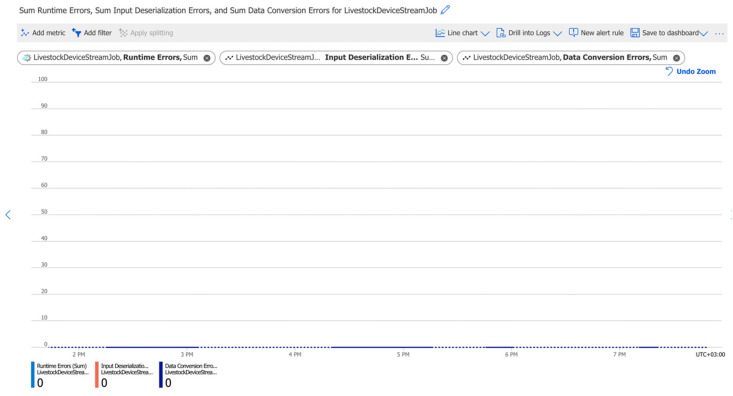
Stream Job sum of errors. The blue line shows the sum of runtime errors. The Red line shows the sum of input errors. The dark blue line shows the sum of data conversion errors. The dotted line shows that no data is ingested for that period (when IoT devices are in sleep mode).

**Figure 12 sensors-22-06566-f012:**
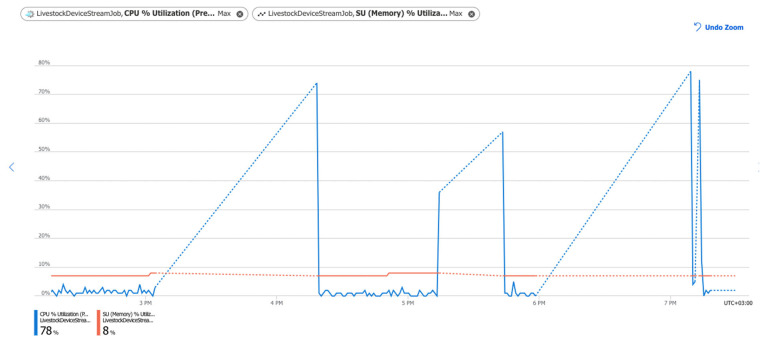
Stream Job Utilization Percentage. The blue line shows the percentage of CPU Utilization. The Red line shows the percentage of SU Utilization. The dotted line shows that no data is ingested for that period (when IoT devices are in sleep mode).

**Figure 13 sensors-22-06566-f013:**
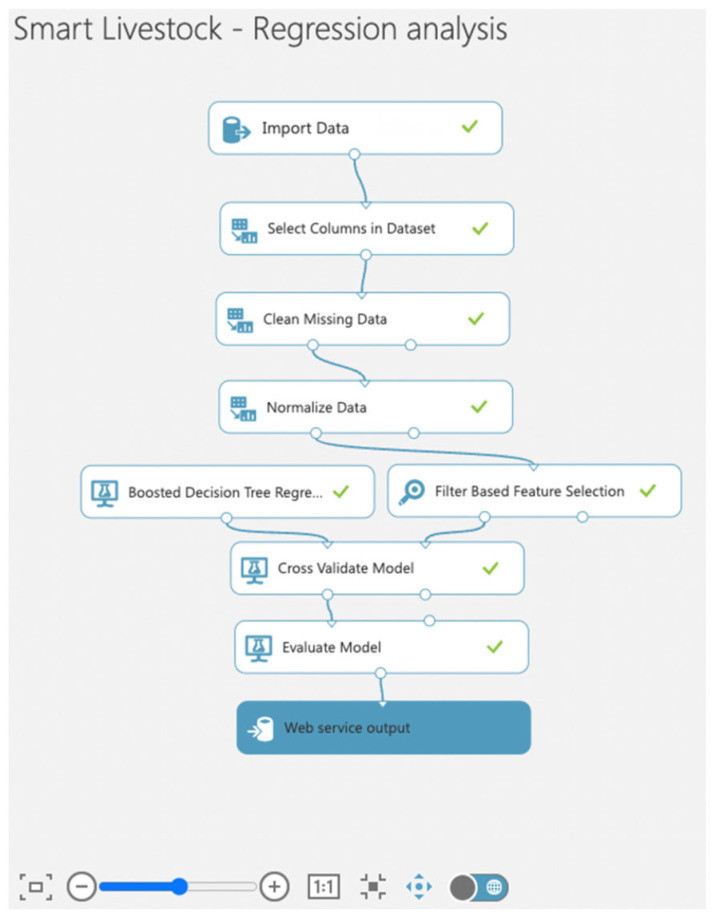
Azure Machine Learning workflow.

**Figure 14 sensors-22-06566-f014:**
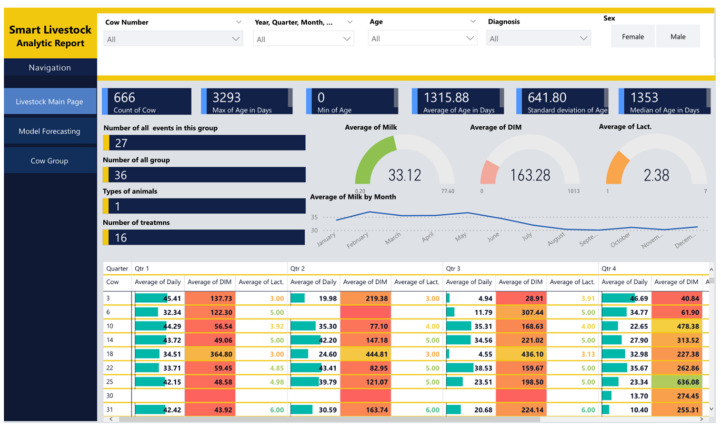
Smart Livestock monitoring system—Analytic report. The data table uses “heat rules” that are restricted to two colors and a gradient in-between. Values above the column average have a green background. Values below column average have a red background. Values close to column average have a gradient background.

**Figure 15 sensors-22-06566-f015:**
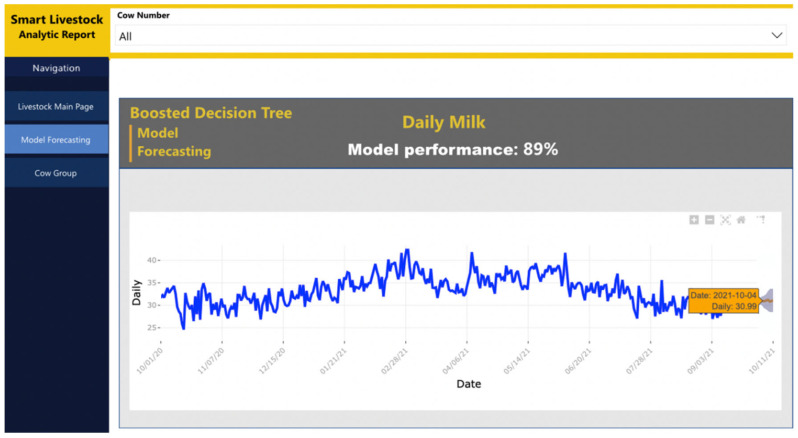
Machine learning model visualization.

**Figure 16 sensors-22-06566-f016:**
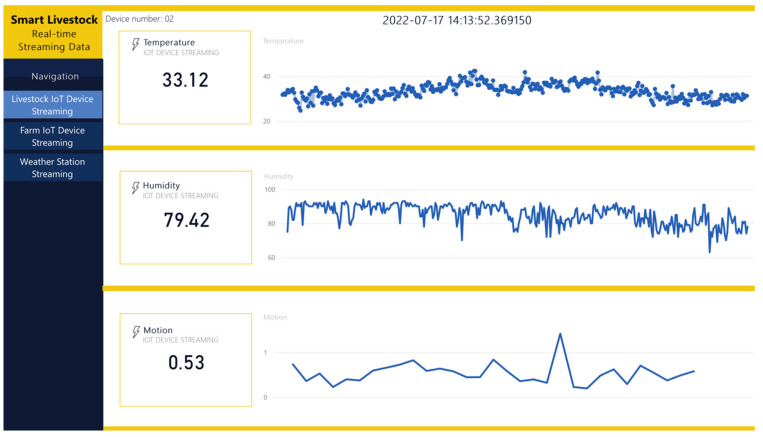
Real-time report with data from livestock IoT device.

**Figure 17 sensors-22-06566-f017:**
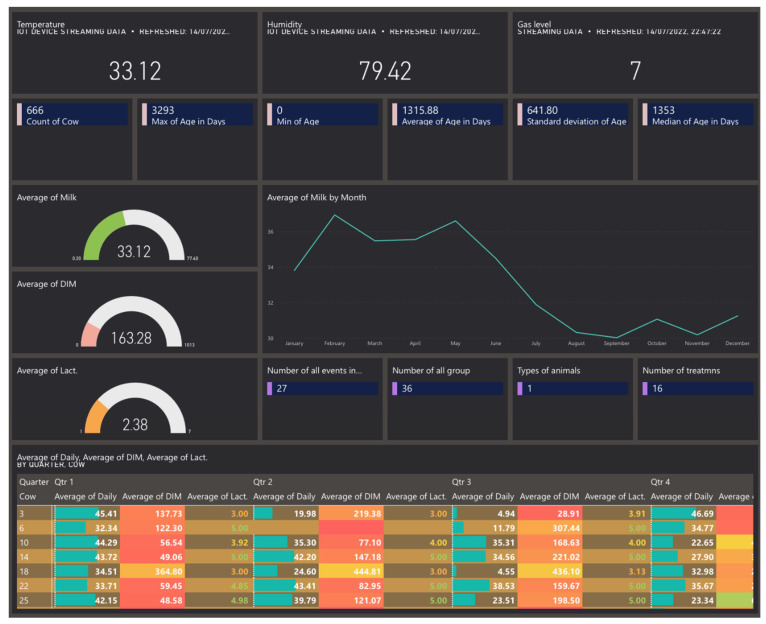
Smart Livestock monitoring system—Dynamic Dashboard. The data table uses “heat rules” that are restricted to two colors and a gradient in-between. Values above the column average have a green background. Values below column average have a red background. Values close to column average have a gradient background.

**Figure 18 sensors-22-06566-f018:**
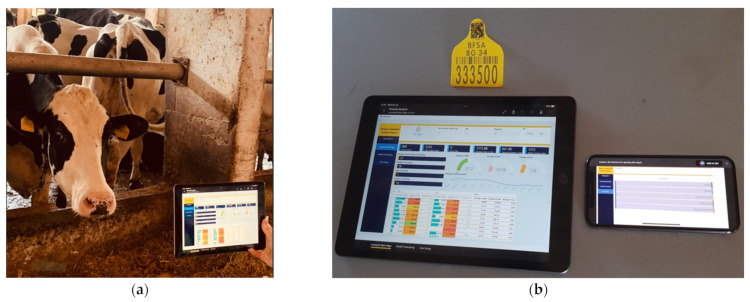
(**a**) Scanned ear tag with lasered QR code—testing in a real environment, and (**b**) scanned QR tag by different devices.

## Data Availability

Not applicable.

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
