# Peer review of "Cloud Data-Driven Intelligent Monitoring System for Interactive Smart Farming"

_sensors, 2022, doi:10.3390/s22176566_

Round 1
Reviewer 1 Report
This manuscript proposed an intelligent monitoring system for livestock smart farming using Microsoft Azure Cloud. In fact, the smart farming system using a cloud is not a new concept. There have already been many proposals, and commercialized smart farm management systems using the cloud, such as Priva Cloud, already exist. Therefore, the main contribution of this manuscript seems to be the implementation of an intelligent monitoring system for a real smart farm using a commercial cloud (Microsoft Azure Cloud).
The manuscript was basically easy to read and understand. However, some information should be added to contribute to this manuscript.
1. There are other commercial clouds, such as Amazon AWS. Thus, the authors should explain why Microsoft Azure Cloud is selected.
2. It would be nice if an explanation of privacy and security, one of the challenges, was added. For example, why privacy and security are important in smart farms
3. Some figures, such as figures 4 to 13, are relatively small and of poor quality. In particular, the size of the text is too small compared to the size of the figure, which makes it difficult to read.
4. Figure 11 shows that there are no stream job errors. Of course, error-free is good, but simply showing that there are no errors doesn't give the reader any insight. Therefore, it is questionable whether this figure should have been included in the manuscript. Rather, it is likely to give the reader a lot of insight by showing examples of real errors in this figure and explaining why these errors occurred and how to fix them.
5. Azure Cloud supports various preprocessing methods and machine-learning algorithms. Thus, the authors should explain why they choose the preprocessing method and machine-learning algorithm in the manuscript.
6. It would be nice if there was an explanation of which features among the selected features are highly correlated with the milk production.
7. It would be nice to add explanations such as how often the ready-to-use model is updated, whether to learn all data collected so far or only the latest data after a certain period of time when the model is updated, and how long it takes to create the updated model.
8. The author mentioned the limitations of the system, but it would be good if they also explain the cost of using Azure Cloud.
9. The intelligent monitoring system proposed in this paper has already been used in two livestock smart farms. Therefore, it is likely that the feedback received through a survey on the proposed system for real users can better represent the contribution of this paper.
10. There are some typos.
A. 4.296 à 4,296 (4 page)
B. humidity (PH) à humidity (RH). In Figure 1b, it seems that humidity is expressed in RH (relative humidity). (4 page)
C. 200.000 à 200,000 (14 page)
Author Response
The authors would like to sincerely thank the reviewers for their time to review the submitted manuscript and for the provision of valuable comments.
We did our very best to address them in the revised version of the manuscript.
Revisions in the text related to the reviewers’ comments are in red.
Review 1
This manuscript proposed an intelligent monitoring system for livestock smart farming using Microsoft Azure Cloud. In fact, the smart farming system using a cloud is not a new concept. There have already been many proposals, and commercialized smart farm management systems using the cloud, such as Priva Cloud, already exist. Therefore, the main contribution of this manuscript seems to be the implementation of an intelligent monitoring system for a real smart farm using a commercial cloud (Microsoft Azure Cloud).
Point 1. There are other commercial clouds, such as Amazon AWS. Thus, the authors should explain why Microsoft Azure Cloud is selected.
Response 1:
The authors would like to thank the reviewer for his observations. Changes are done in the manuscript.
The authors have been using various cloud platforms in their studies. Based on the experience, we preferred Azure as the cloud provider for the proposed IoT systems.
The authors work on another article for a comparison between various cloud providers. They will compare the types of problems that occurred during development and operation, their frequency, and reasons for occurrence, observed latency by periods, funds spent by months, the services maintenance complexity, security, and many others.
Added in Manuscript in Section 1: Introduction
Microsoft Azure is used in the design and implementation process of the proposed system. Azure is a cloud computing platform with a vast ecosystem of managed cloud services and resources such as VPC, storage, databases, networking, software, analytics, and intelligence.
According to the Gartner report, Microsoft is the market leader with Azure IoT, a set of services delivered on the Azure platform, among 18 others considered cloud providers. The main strengths of the platform are flexible IoT, Al/ML solution deployment approach, strong security approach, global partner ecosystem and more.
Point 2. It would be nice if an explanation of privacy and security, one of the challenges, was added. For example, why privacy and security are important in smart farms.
Response 2:
Thank you for the question regarding security. The authors consider that security is important not only for IoT-based systems but also for all Internet-based systems. The concerned justifications were added to the manuscript.
Added in Manuscript in Section 1: Introduction
- Privacy and security. In a smart farming ecosystem, there are four classes of cyber-attacks – data attacks, networking and equipment attacks, supply chain and other relevant attacks. Therefore, all entry-exit points of the system must be doubly protected.
Added in Manuscript in Section 2.3: Smart Livestock system architecture
IoT Hub is the only external point for the system. Part of IoT Hub is Defender for IoT service that provides comprehensive monitoring and alerts in case of threat detection against IoT environments. It is a standalone service designed to add an additional layer of threat protection for IoT Hub and registered IoT devices. In addition, IoT Defender uses masked IP addresses and processes and stores the data in a different geographical location than the IoT Hub.
Added in Manuscript in Section 4: Discussion
Privacy and security
The use of IoT devices and intelligent communication technologies increases cybersecurity threats and vulnerabilities in smart farming environments that have specific attributes such as farm equipment, labour sharing, and operational approaches and decisions.
Collected farm-specific data such as farm location, yields, number of animals and their conditions, available quantities of products in the warehouse, equipment, staff shifts, and more are of interest to competitors, distributors, prospective investors and others with malicious intent.
Therefore, the development of intelligent farm systems requires more attention and a high level of security The proposed system is developed in a cloud environment due to the high level of security present at every layer.
IoT defender is an additional service used for an additional level of security for both IoT Hub and individual IoT devices. This way, when a physical attack is performed on an IoT device, the attacker will not receive any sensitive information, and the owner will be immediately notified.
Gaining control over insecure IoT systems also can be used later for launching large-scale DoS attacks against other internet-based systems causing huge damages.
Such cyber-attacks have the potential to disrupt the economies of countries that are heavily dependent on smart agriculture.
Point 3. Some figures, such as figures 4 to 13, are relatively small and of poor quality. In particular, the size of the text is too small compared to the size of the figure, which makes it difficult to read.
Response 3:
The authors fully accept the remark and apologies for the inconvenience.
Following MDPI Sensors – Instructions for authors, section “Preparing figures, schemes and tables”, the authors will provide to the editors all figures in full size and quality (minimum 1000 pixels width/height, or a resolution of 300 dpi or higher) in a single zip archive. This will allow the readers to see the original quality of all figures.
Point 4. Figure 11 shows that there are no stream job errors. Of course, error-free is good, but simply showing that there are no errors doesn't give the reader any insight. Therefore, it is questionable whether this figure should have been included in the manuscript. Rather, it is likely to give the reader a lot of insight by showing examples of real errors in this figure and explaining why these errors occurred and how to fix them.
Response 4:
Thank you for noticing the absence of errors in the system.
The authors believe that providing figures showing error diagrams is useful as it indicates to the user that error monitoring is available in the proposed system. Error handling is a major concern for any system design and development. Showing errors allows for better transparency and reassurance for the end user. The authors experienced neither infrastructure errors nor data-related errors.
Added in Manuscript in Section 3.1: Data-driven pipelines
The chart shows the lack of errors for a short period of time.
However, the chart can be useful for easily spotting errors in long-term series if they occur somewhere along the pipeline. In general, infrastructure-related errors are unlikely to be expected as cloud providers are well-known and ensure that they provide services with high levels of durability, reliability, and availability.
Data conversion errors can occur when messages received into the system are incorrectly formatted. The structure of the message is controlled by the logic implemented in the IoT device. All messages have a uniform interface and are sent to the system as JSON objects.
Point 5. Azure Cloud supports various preprocessing methods and machine-learning algorithms. Thus, the authors should explain why they choose the preprocessing method and machine-learning algorithm in the manuscript.
Response 5:
The article presents the final version of the developed Azure Machine Learning workflow which showed the best results among others. This was achieved after performing numerous tests and experiments conducted in a testing workspace dedicated to the study, during which various data pre-processing steps and ML algorithms were tested and several ML models were constructed. Testing of the developed models proved the applicability of the developed cloud-based workflow, with the Boosted Decision Tree Regression Model showing the highest accuracy in predicting the amount of milk. The results of these experiments are detailed in another paper of ours, which is currently in print.
The research we present in this article is mainly focused on the design and implementation of an end-to-end system for receiving and reporting data. That is why the pre-processing method and machine learning algorithm are only briefly described.
Added in Manuscript in Section 3.2: Azure Machine Learning
...MICE is considered due to data heterogeneity and seasonality. Using other available methods such as mean replacement, median, mode and others introduces noise or significantly reduces the total number of observations used.
… Z-score is the chosen method for data normalization because it handles outliers better.
Unlike other methods such as Standard Scaler and Min-Max, the Z-score indicates how many standard deviation units an individual observation is away from the mean. It calculates in confidence interval [-3: 3] [36] (p. 64).
...
The fact that z-scores belong to the standard normal distribution makes it possible to use z-scores to compare heterogeneous values of primary measurements.
Pearson correlation…
This method was chosen over its alternatives such as the Spearman and Kendal correlation because the data were normally distributed and the prediction of numerical but not categorical results. Furthermore, this method is not affected by changes in the scale of the variables used.
...
The ready-transformed data is trained and tested with Boosted Decision Tree algorithm. Boosted Decision Tree Regression is an ML algorithm for solving regression tasks, that creates a prediction model in the form of an ensemble of decision trees, thereby allowing the prediction of the value of the target variable by learning simple decision rules, inferred from data features. A decision tree is a structure in which each internal node represents a "test" of an attribute; each branch represents the result of a test. It is suitable for datasets when there are multiple training features.
...
Added in Manuscript in Section 4: Discussion
Azure Machine Learning Studio is the selected tool for developing ML models. It is provided by Azure as a software as a service (SaaS). This is an important feature because Azure takes care of provisioning and maintaining compute, storage, and global infrastructure.
As a result of the applied approach presented in figure 13, the ingested data is cleaned, transformed, and normalized. A regression model was created to predict future milk quantities. The model is implemented as a web service. The performance of the model is monitored to detect problems with the machine learning model, such as overfitting and underfitting.
Point 6. It would be nice if there was an explanation of which features among the selected features are highly correlated with the milk production.
Response 6:
The reviewer correctly observed that certain parameters were highly correlated with the output variable. We have added relevant comments to the text.
Added in Manuscript in Section 3.2: Azure Machine Learning
As a result of this step, twelve features were outlined that had the strongest predictive power and could be used to predict the amount of milk produced. These are Lactation (the period between one calving and the next), Age, DIM (days in milk), Dry (a stage of their lactation cycle where milk production ceases prior to calving), Season, Humidity, Temperature, Breed, Animal Noise Level, Motion, Days in Group and BCS (body condition scoring).
Point 7. It would be nice to add explanations such as how often the ready-to-use model is updated, whether to learn all data collected so far or only the latest data after a certain period of time when the model is updated, and how long it takes to create the updated model.
Response 7:
Thank you very much for the question. Explanations are provided in the manuscript.
Added in Manuscript in Section 3.2: Azure Machine Learning
There are some events that can trigger model updates.
The first one is if the performance of the model starts to deteriorate at some pre-determined threshold. The process of retraining and fine-tuning the model is then triggered. Another one if the data schema is changed and new features are introduced into the model.
The model is retrained with the new data and the historical data before the model is deployed into production. The difference is that the already known model parameters are used as a starting point during the retraining process. This significantly reduces the time to create a production-ready machine learning model.
Point 8. The author mentioned the limitations of the system, but it would be good if they also explain the cost of using Azure Cloud.
Response 8:
According to your note, we have added the necessary explanations.
Added in Manuscript in Section 4: Discussion
System Limitations and costs
Azure Cloud works according to the OpEx (Operating Expenses) model, which allows the system to scale up or down to meet the specific capacity needs. Unlike the CapEx (Capital Expenditures) model (which is used for on-premises systems), where upfront cost on physical infrastructure and expenses planning is required at the start of the project, in the OpEx model resource usage is charged based on the usage following the "pay-as-you-go' model. One major benefit of OpEx model is that there are no wasted resources because if resource usage is not needed, then it is not provisioned and subsequently not paid.
The system uses standard pricing tier (S1):
IoT Hub - $25 monthly per IoT Hub unit.
Defender for IoT - $0,001 monthly per connected device.
Stream Analytics - $1 monthly per device.
Blob storage (Hot) - $0,02 monthly per GB for first 50 terabyte (TB).
Azure Machine learning - $9,99 monthly per ML studio workspace and $1 per studio experimentation hour.
Power BI Pro - $9,99 monthly per user.
Point 9. The intelligent monitoring system proposed in this paper has already been used in two livestock smart farms. Therefore, it is likely that the feedback received through a survey on the proposed system for real users can better represent the contribution of this paper.
Response 9:
With the launch of the National Scientific Program “Intelligent Animal Husbandry”, numerous discussions were held with farmers and veterinarians to explore the current state of the sector and the desired future technological improvement.
Periodically, the intermediate results of the development of the implementation of the Program were presented and the corresponding corrections were made. It is a long and complicated process because each participant has different ideas, views, understandings, and perceptions
Numerous alpha-beta tests have been performed on the dashboard reports and visualizations to greatly improve the user experience. The design and structuring work of the visuals was done with farmers' satisfaction in mind. After presenting the results, farmers and veterinarians and already started asking for new features and functionalities to be added. Future work may be outlined with the following text added to the manuscript.
Added in Manuscript in Section 4: Discussion
Future work
The system is planned to be improved and enriched with new functionalities such as:
- Research and design of new features for ML models.
- Research and development of approaches, techniques and tools to automatically calculate BCS (body condition score) to better track the health status and fat levels of the animal along with the general condition of the herd.
- Research on automatic analysis of daily animal activity to improve early detection of animal diseases.
- Designing and building additional reports and dashboards for monitoring and improving their interactivity.
- Automatic generation and export of daily, weekly, monthly, and annual reports with statistical information, both for the entire farm and for each individual animal. Automatic comparative analysis against generated results.
After the future development of the functionalities, it will be necessary to add new services and increase the capacity of the system.
Point 10. There are some typos.
A. 4.296à4,296 (4 page)
B. humidity (PH)àhumidity (RH). In Figure 1b, it seems that humidity is expressed in RH (relative humidity). (4 page)
C. 200.000 à 200,000 (14 page)
Response 10:
Thank you for noticing this. Necessary corrections have been made.
Note: We attach the edited manuscript.
The authors have used the "Red colour" in the article document to integrate all the changes indicated by the reviewer. We remain attentive to your feedback and are very grateful for your valuable help.
The authors are grateful for valuable and constructive comments. We believe that the changes made have increased the impact of the proposed article.

Reviewer 2 Report
The article is devoted to the problem of applied automation. The structure of the article is classical. The article is easy to read. The level of English is acceptable. The quality of the figures varies. For example, to make out what is written in small print in Fig. 9-14 is impossible. The article cites 38 relevant sources.
The following remarks can be made on the material of the article:
1. Formally, the authors have an article, since the material in all the required sections is available. In fact, the material given in the flock does not meet the required level for scientific articles. Let's analyze the topic of the article sequentially. The topic of the article mentions "intellectuality". In the Models and Methods section, it is only declared. No design, implementation and optimization of an intelligent control system.
2. In section 2.3, a conveyor is declared. It's fine. You immediately expect to see a harmonious mathematical apparatus of the theory of queuing. But this is not about this article. Instead, we see Fig. 6, in the name of which the conveyor is declared, but in fact it is absent. The figure is followed by lengthy comments.
3. The Experiments section should begin with a description of statistically representative test data. This is not. Here, UML diagrams of activity, sequence and deployment would be appropriate, and not Fig. 7, 8.
4. The Discussion section is written with a process engineer in mind. This is fine for a handout, but not for a scientific publication. However, this remark applies to the entire article. In this form, the article is a good report for an engineering project, but not a scientific work.
Author Response
The authors would like to sincerely thank the reviewers for their time to review the submitted manuscript and for the provision of valuable comments.
We did our very best to address them in the revised version of the manuscript.
Responses in the text related to the reviewers’ comments are in blue.
Review 2
The article is devoted to the problem of applied automation. The structure of the article is classical. The article is easy to read. The level of English is acceptable. The quality of the figures varies. For example, to make out what is written in small print in Fig. 9-14 is impossible. The article cites 38 relevant sources.
The authors accept the remark in full and apologize for any inconvenience.
Following MDPI Sensors – Instructions for authors, section “Preparing figures, schemes and tables”, the authors will provide to the editors all figures in full size and quality (minimum 1000 pixels width/height, or a resolution of 300 dpi or higher) in a single zip archive. This will allow the readers to see the original quality of all figures.
Point 1. Formally, the authors have an article, since the material in all the required sections is available. In fact, the material given in the flock does not meet the required level for scientific articles. Let's analyze the topic of the article sequentially. The topic of the article mentions "intellectuality". In the Models and Methods section, it is only declared. No design, implementation and optimization of an intelligent control system.
Response 1:
The authors would like to thank the reviewer for the valuable note. Changes have been made in the manuscript. Additional references are included to enhance the arguments of the article.
Added in Manuscript in (Section 1: Introduction)
The trend of bringing more intelligence to farm monitoring and control systems is widely observed. An intelligent system (IS) can be defined as a system that incorporates intelligence into machine-processed applications. It is an advanced system that can collect, analyze, store, and respond to the data it gathers from the environment. It can operate and communicate with users or other computer systems. It can also learn from experience and adapt according to current data.
Advances in digital technology are leading to the development of intelligent systems that can monitor, control, and visualize various farm operations and animal status in real-time. The wireless sensor network and cloud application software management system based on the Internet of Things (IoT) is used to design an intelligent monitoring system for the facility agriculture environment. An intelligent monitoring system which is based on machine vision, an automatic identification model, a Web client, and a cloud server is developed in [15].
Added in Manuscript in (Section 2.3: Data-driven system architecture)
Intelligent systems include a flow of behavior between hardware and software components. The design of intelligent systems includes some key characteristics:
- Sensors – collects data from the environment and transmit it to the system core for identification and analytics.
- Identification – intelligent systems must automatically recognize specific information and transmit it to subscribed services or channels.
- Data analytics - an essential function of an intelligent system is its ability to process collected data.
- Self-learning - the intelligent system needs to include artificial intelligence or machine learning functionalities.
- Real-time communication - an intelligent system needs to have the ability to simulate or emulate in real-time or near real-time.
- User Experience (UX) - to interact with users, intelligent systems must have interfaces such as web pages, reports, dashboards, or other types of visualization.
- Remote system management - an intelligent system allows users to interact with it from any location.
- Interoperability and connectivity – an intelligent system must combine its elements into a holistic communication process.
- Protection – An intelligent system's ecosystem, networks and communications must be secure to be available and reliable to function properly.
Point 2. In section 2.3, a conveyor is declared. It's fine. You immediately expect to see a harmonious mathematical apparatus of the theory of queuing. But this is not about this article. Instead, we see Fig. 6, in the name of which the conveyor is declared, but in fact it is absent. The figure is followed by lengthy comments.
Response 2:
The authors apologise that the diagram title misled the reviewer.
In fact, this figure shows the authors' approach to creating a monitoring system. The selection of services, the communication and interaction of the service and all necessary settings are not a commercial approach.
The authors aim to confirm that the proposed approach is suitable for the intended purposes. Since the system is entirely Cloud-based, it is important that the built approach considers Cloud-native techniques during system design and implementation. The authors' experience in developing such types of systems helps to choose the best set of cloud services among many others provided by Azure to build a ready-to-use, highly automated system that is infrastructure independent and does not need large initial capital investment and maintenance manpower.
Edited in Manuscript in (Section 2.3: Data-driven system pipeline -> Data-driven system architecture)
Figure 6. Smart Livestock data-driven pipeline -> Smart Livestock system architecture
Added in Manuscript in (Section 2.3: Data-driven system architecture)
Figure 6 shows the developed architecture of the proposed intelligent monitoring system which includes the key characteristics for the design of an intelligent system. It consists of an IoT part, various cloud services and connections between them. Implementing the entire process from data extraction through storage, filtering, processing, forecasting, and visualization is very complex and requires a thorough selection of cloud services and the ways of communication between them.
…
During the development process of the built cloud architecture, the five pillars of the well-architected systems were followed. These are reliability, security, cost optimization, operational excellence, and performance efficiency. They describe the design principles and the best architectural practices for designing and executing cloud workloads. Following these pillars helps produce a high-quality, stable, reliable, scalable, and efficient cloud-based system.
Point 3. The Experiments section should begin with a description of statistically representative test data. This is not. Here, UML diagrams of activity, sequence and deployment would be appropriate, and not Fig. 7, 8.
Response 3:
The authors propose a data-driven monitoring system with machine learning capabilities. For this reason, the experimental section begins with system details and features. Following your comment, changes were made to the manuscript and the data representation is described in the machine learning section where the data processing is explained.
The authors believe that UML provides some useful value for visualizing objects, class relationships, and behavioural system characteristics in a standardized way. Figures 7 and 8 were created with Azure native tools with additional labels and important real-time metrics included. Azure will automatically update the figures accordingly if subsequent changes are introduced, which is not the case with creating UML diagrams where someone needs to keep them in sync. In addition, the authors try to use as many cloud-native tools as possible. Figures 7 and 8 are native to the Azure ecosystem, and any Azure user will easily understand the semantics these figures represent.
Added in Manuscript in (Section 3.2 Azure Machine Learning)
The data consists of observations for a period of over 2 years. It has in total 2,285,665 values organized in 43 columns and 53,155 rows. Data is in a raw format and has several data types like datetime64[ns] (4 columns), float64 (13 columns), int64(7 columns), and object (19 columns). 11% out of the values are null values.
…
“Daily” is the target variable for prediction representing a daily quantity of milk per cow. Statistical characteristics are as follows: mean: 33.115; standard deviation: 13.113; minimum: 0.00; 25%: 24.600; 50%: 33.700; 75%: 42.400; max 77.400; 70% non-null values.
Point 4. The Discussion section is written with a process engineer in mind. This is fine for a handout, but not for a scientific publication. However, this remark applies to the entire article. In this form, the article is a good report for an engineering project, but not a scientific work.
Response 4:
The authors acknowledge the reviewer's comments. Changes are made throughout the manuscript.
Note: We attach the edited manuscript.
The authors have used "blue colour" in the article document to integrate all changes indicated by the reviewer. We remain attentive to your feedback and are very grateful for your valuable help.

Round 2
Reviewer 1 Report
The authors have satisfactorily addressed the Reviewer's concerns.
Author Response
The authors would like to thank the reviewers for their time to review the submitted manuscript and for the provision of valuable comments.
You can find the latest changes in the manuscript as an attachment.

Reviewer 2 Report
I made the following remarks to the basic version of the article:
1. Formally, the authors have an article, since the material in all the required sections is available. In fact, the material given in the flock does not meet the required level for scientific articles. Let's analyze the topic of the article sequentially. The topic of the article mentions "intellectuality". In the Models and Methods section, it is only declared. No design, implementation and optimization of an intelligent control system.
2. In section 2.3, a conveyor is declared. It's fine. You immediately expect to see a harmonious mathematical apparatus of the theory of queuing. But this is not about this article. Instead, we see Fig. 6, in the name of which the conveyor is declared, but in fact it is absent. The figure is
3. The Experiments section should begin with a description of statistically representative test data. This is not. Here, UML diagrams of activity, sequence and deployment would be appropriate, and not Fig. 7, 8.
4. The Discussion section is written with a process engineer in mind. This is fine for a handout, but not for a scientific publication. However, this remark applies to the entire article. In this form, the article is a good report for an engineering project, but not a scientific work.
The authors responded to my comments. Unfortunately, their answers correspond to the level of the article - they are bright, general and not analytically substantiated. The article resembles a bachelor's work done on a Microsoft engine. Is such material worthy of being published in a Q1 journal? The final decision is made by the editor.
Author Response
The authors would like to thank the reviewer for their time to review the submitted manuscript.
You can find the latest changes in the manuscript as an attachment.
